# Stigma, depression, quality of life, and the need for psychosocial support among people with tuberculosis in Indonesia: A multi-site cross-sectional study

**Ahmad Fuady**[1,2], **Bustanul Arifin**[3,4], **Ferdiana Yunita**[5], **Saidah Rauf**[6], **Agus Fitriangga**[7], **Agus Sugiharto**[1], **Finny Fitry Yani**[8,9], **Helmi Suryani Nasution**[10], **I. Wayan Gede Artawan Eka Putra**[11], **Muchtaruddin Mansyur**[1], **Tom Wingfield**[12,13,14]*

1 Department of Community Medicine, Faculty of Medicine, Universitas Indonesia, Jakarta, Indonesia, 2 Primary Health Care Research and Innovation Center, Indonesian Medical Education and Research Institute, Faculty of Medicine Universitas Indonesia, Jakarta, Indonesia, 3 Faculty of Pharmacy, Universitas Hasanuddin, Makassar, Sulawesi Selatan, Indonesia, 4 Unit of Global Health, Department of Health Sciences, University of Groningen, University Medical Centre Groningen (UMCG), Groningen, The Netherlands, 5 Department of Community Medicine, Faculty of Medicine, Universitas Gunadarma, Depok, Indonesia, 6 Department of Nursing, Politeknik Kesehatan Kemenkes Ambon, Maluku, Indonesia, 7 Department of Community Medicine, Faculty of Medicine, Universitas Tanjungpura, Pontianak, West Kalimantan, Indonesia, 8 Department of Child Health, Faculty of Medicine, Universitas Andalas, Padang, West Sumatera, Indonesia, 9 Department of Paediatric, Dr. M. Djamil General Hospital, Padang, West Sumatera, Indonesia, 10 Department of Public Health, Faculty of Medicine and Health Sciences, Universitas Jambi, Jambi, Indonesia, 11 Department of Public Health and Prevention Medicine, Faculty of Medicine, Universitas Udayana, Kota Denpasar, Bali, Indonesia, 12 Department of Clinical Sciences and International Public Health, Liverpool School of Tropical Medicine, Liverpool, United Kingdom, 13 Department of Global Public Health, WHO Collaborating Centre on Tuberculosis and Social Medicine, Karolinska Institute, Stockholm, Sweden, 14 Tropical and Infectious Disease Unit, Royal Liverpool and Broadgreen University Hospitals NHS Trust, Liverpool, Liverpool, United Kingdom

* tom.wingfield@lstmed.ac.uk

## Abstract

Stigma towards people with tuberculosis (TB-Stigma) is associated with other psychosocial consequences of TB including mental illness and reduced quality of life (QoL). We evaluated TB-Stigma, depression, QoL, and the need for psychosocial support among adults with TB in Indonesia, a high TB burden country. In this primary health facility-based survey in seven provinces of Indonesia, from February to November 2022, we interviewed adults receiving (a) intensive phase treatment for drug-susceptible (DS) TB at public facilities, (b) treatment at private facilities, (c) those lost to follow up (LTFU) to treatment, and (d) those receiving TB retreatment. We used our previously validated Indonesian TB-Stigma Scale, Patient Health Questionnaire-9, and EQ-5D-5L to measure TB-Stigma, depression, and QoL. Additional questions assessed what psychosocial support was received or needed by participants. We recruited and interviewed 612 people, of whom 60.6% (96%CI 59.6–64.5%) experienced moderate TB-Stigma. The average TB-Stigma scores were 19.0 (SD 6.9; min-max 0–50; Form A-Patient Perspective) and 23.4 (SD 8.4, min-max 0–50; Form B-Community Perspective). The scores were higher among people receiving treatment at private facilities (adjusted B [aB] 2.48; 0.94–4.03), those LTFU (aB 2.86; 0.85–4.87), males (aB 1.73; 0.59–2.87), those losing or changing job due to TB (aB 2.09; 0.31–3.88) and those living in a

**Data Availability Statement:** All relevant data are within the paper and its Supporting Information files.

**Funding:** This was work was supported by grants to AF from the Royal Society of Tropical Medicine (Grant No. 19590206), UK, and Universitas Indonesia (PUTI Q1 Grant No. NKB-1103), Indonesia. This work was supported by grants to TW from the Wellcome Trust (209075/Z/17/Z), UK, the Medical Research Council, Department for International Development, and Wellcome Trust (Joint Global Health Trials, MR/V004832/1), and the Medical Research Foundation (Dorothy Temple Cross International Collaboration Research Grant, MRF-131-0006-RG-KHOS-C0942), UK. This work was supported by a shared Public Health Intervention Development award to both AF and TW from the Medical Research Council (PHIND, MR/Y503216/1). The funders had no role in study design, data collection and analysis, decision to publish, or preparation of the manuscript. The study team is grateful for this funding.

**Competing interests:** The authors have declared that no competing interests exist.

rural area (aB 1.41; 0.19–2.63). Depression was identified in 41.5% (95% CI 37.7–45.3%) of participants. Experiencing TB-Stigma was associated with moderately severe to severe depression (adjusted odds ratio [aOR] 1.23; 1.15–1.32) and both stigma and depression were associated with lower QoL (aB -0.013; [-0.016]-[-0.010]). Informational (20.8%), emotional (25.9%) and instrumental (10.6%) support received from peers or peer-groups was limited, and unmet need for such support was high. There is a sizeable and intersecting burden of TB-Stigma and depression among adults with TB in Indonesia, which is associated with lower QoL. Participants reported a substantial unmet need for psychosocial support including peer-led mutual support groups. A community-based peer-led psychosocial support intervention is critical to defray the psychosocial impact of TB in Indonesia.

## Introduction

Tuberculosis (TB) remains a major global public health challenge leading to 1.6 million deaths in 2021 [1]. The World Health Organization's 2015 End TB Strategy called for worldwide TB elimination [2]. However, achieving key Strategy targets including reducing TB incidence and mortality, and eradicating catastrophic costs, is complex and has been challenged further in recent years by the COVID-19 pandemic.

TB is well recognized as a social infectious disease with determinants including poverty, limited healthcare access, and—the often overlooked—stigma [3, 4]. People with TB, their families, and even social networks face persistent stigma from others including in the neighborhood and workplace [5]. Stigma is deeply rooted in myths, misconceptions, misunderstanding, and erroneous health beliefs concerning TB disease and those affected [6, 7]. Such stigma is pernicious and associated with social discrediting and profound feelings of "shame, self-rejection, and self-loathing" amongst those affected [8].

Stigma towards and experienced by people with TB, herein termed TB-Stigma, has been recognized by the Global Fund [9] and United Nations [10] as a global public health challenge and a critical barrier to achieving the World Health Organization (WHO) goal of ending TB by 2050 [11]. TB-Stigma is classified into enacted or experienced, anticipated, and internalized or self-stigma, all of which complicate efforts to control and eliminate TB [12, 13]. Enacted and anticipated stigma, ranging from expectation and fear of discrimination to experiences of stigmatizing behaviour by others, can prevent people with TB from disclosing their symptoms and signs or seeking care and being identified for testing [14]. This can compound delays in healthcare seeking and TB diagnosis [15] and reduce treatment adherence and success [16]. TB-Stigma among people with TB, including in both formal and informal work sectors, may also lead to income loss and catastrophic health expenditure [17]. Enacted and internalized stigma and the economic consequences of each can precipitate or aggravate mental illness [18] and reduced quality of life (QoL) [19, 20].

There is a substantial burden of TB-Stigma in low- and middle-income countries (LMICs), particularly with high TB burden [19, 21, 22]. Indonesia is a LMIC with 969,000 TB cases annually, contributing 9.2% of the total estimated TB incidence worldwide [1]. Despite these stark figures and the recognition of the vital role that addressing TB-Stigma will play in eliminating TB, there is limited evidence concerning TB-Stigma and its impact in Indonesia [23]. Of the handful of qualitative studies exploring stigma among Indonesian people with TB [24–26], there has been negligible concomitant assessment of associated symptoms of depression,

mental illness, and QoL; and little consideration of potential interventions to mitigate these psychosocial consequences of TB.

For Indonesia to work towards ending TB, it will be essential to measure the "psychosocial" impact and consequences of TB such as TB-Stigma, depression, and QoL; and to identify and address the need for psychosocial support amongst TB-affected people. This much-needed data will support generation of evidence to inform the design and delivery of impactful, sustainable, and locally appropriate psychosocial interventions to reduce TB-Stigma, mitigate mental illness, and improve people with TB's QoL. As part of a larger program of research, this study aimed to fill this knowledge gap in Indonesia by assessing TB-Stigma, depression, QoL and their correlations, as well as measuring the need for psychosocial support among adults with TB in Indonesia. The results of this study will support design of a community-based psychosocial support intervention and influence related national and international TB-Stigma policy and practice.

## Methods

This study was part of the *Characterising and Addressing the Psychosocial Impact of Tuberculosis in Indonesia* (CAPITA) research programme, conducted from February to November 2022 [27]. The study findings will feed into the associated Medical Research Council Public Health Intervention Development-funded (MR/Y503216/1) "TB-CAPS" study to co-design and develop a community-based, peer-led psychosocial support intervention to combat TB-Stigma in Indonesia, which commences in July 2023.

### Setting

CAPITA was a study consisting of a cross-sectional survey of adults with TB at primary health facilities in seven provinces of Indonesia. The provinces were selected purposively to represent areas of diverse geography (western, central, and eastern regions) and TB burden (medium to high) in Indonesia. Since TB-Stigma and TB psychosocial impact may vary between rural and urban areas [28, 29], we also purposively selected one urban city and one rural district in each province, based on Indonesian National Statistical Bureau data and consultation with the National TB Program (NTP) officers at the provincial level (**Fig 1**).

### Participant selection

We interviewed adults aged ≥18 years at either public or private primary healthcare facilities receiving treatment for drug-susceptible (DS) TB free of charge under the coordination of the Indonesian NTP. We reviewed the NTP registers at each health facility and selected participants based on their TB treatment status. A priori, we pragmatically divided the potential participants into four groups to cover the broadest possible representative public-private mixed sample of people with DS-TB in Indonesia. *Group A* was people receiving DS-TB treatment regimen for the first time at a public primary healthcare facility and in the intensive phase (the first two months of a standard six-month DS-TB regimen). *Group A* participants were selected consecutively from the person most recently diagnosed, notified, and registered backwards. *Group B* was people receiving TB treatment at private primary healthcare facilities. *Group C* was people who were diagnosed with TB at public healthcare facilities but never started TB treatment termed "lost to follow up to treatment" (LTFU). Groups B and C are critical groups given evidence suggesting that a significant proportion of people with TB symptoms in Indonesia seek care at private healthcare facilities [30, 31] and that TB-Stigma may lead people with TB to be LTFU and/or seek TB care at private facilities [32]. *Group D* was people receiving a DS-TB retreatment regimen at either a public or private primary healthcare facility between

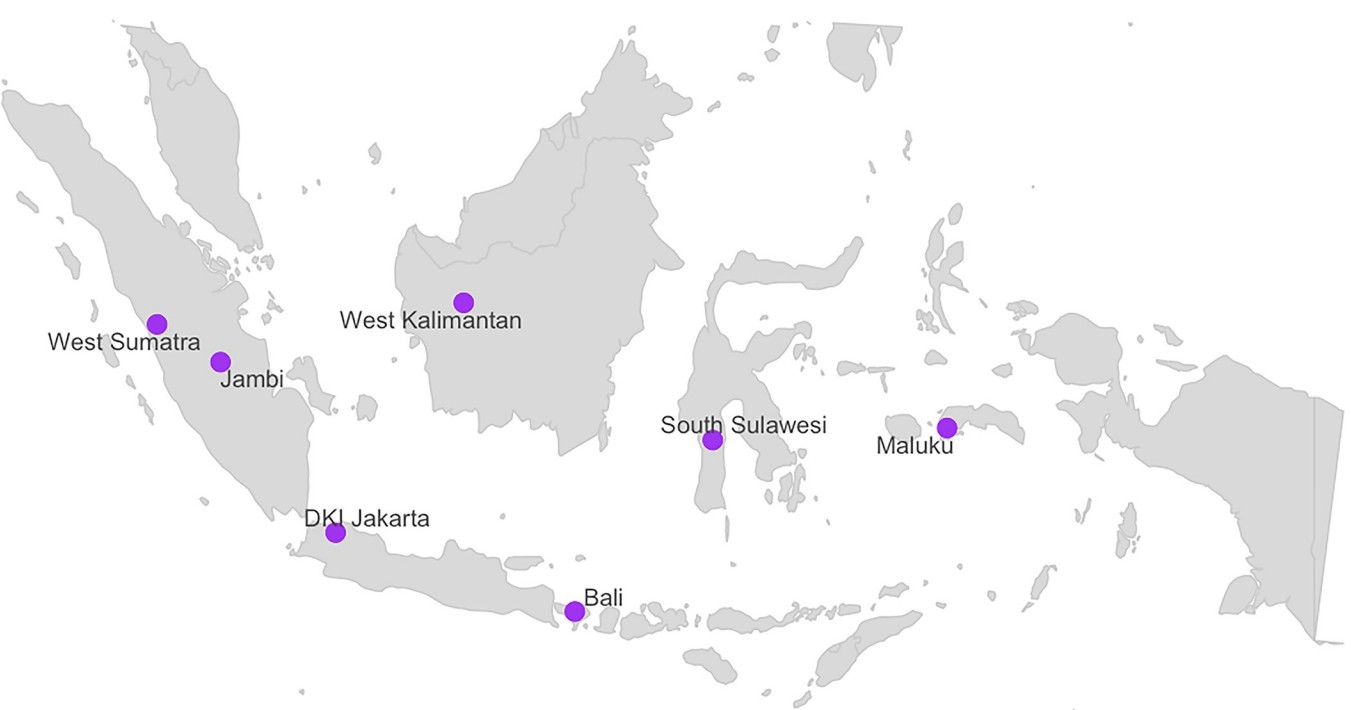

**Fig 1. Selected CAPITA study provinces in Indonesia (purple dots).**

the second week to the ninth month of the retreatment regimen. Group D was selected based on evidence suggesting that TB-Stigma affects people with TB's treatment adherence and can impair TB treatment success rates and increase the likelihood of retreatment, and that there is little evidence on TB-stigma amongst people receiving retreatment for TB [16, 33].

The primary health facility-based survey focused on measuring the "psychosocial" impact and consequences of TB on people receiving DS-TB treatment or retreatment, including TB-Stigma, depression, and QoL. This was supplemented with questions concerning participants' receipt of existing psychosocial support interventions and their perceptions of their unmet need for psychosocial support interventions (described in more detail below). We focused on people with DS-TB and excluded people with drug-resistant TB (DR-TB) because: there have been several studies in Indonesia on the psychosocial consequences for people with DR-TB [34–36]; people with DR-TB receive treatment at a limited number of specific DR-TB treatment primary healthcare facilities with appropriately skilled and trained staff or at secondary-level hospitals; and people with DS-TB contribute >90% of the total number of cases of TB in Indonesia and are at risk of developing acquired resistance if unable to adhere to TB treatment [37]. We also excluded people with DS-TB who had completed TB treatment by the time of recruitment and data collection because previous experience in the field has suggested that the responses of such participants may be unrepresentative or inaccurate due to a high probability of recall bias.

## Sample size

Using data from related, recent research, we assumed that with an average TB-Stigma score of 27.6 (±6.1) with an alpha of 0.05, a precision of 0.7, and urban-rural stratification, we needed at least 584 participants in this study for the findings to be nationally generalisable in Indonesia [14].

## Variables and instruments

The key co-primary outcome variables informing estimates of psychosocial impact were TB-Stigma, symptoms of depression, QoL, and receipt of and/or unmet need for psychosocial support. As described further below, the tools used to measure TB stigma, depression, and QoL have all been previously validated for reliability and internal consistency in the Indonesian context.

### TB stigma scale

We first created, culturally adapted, and validated an Indonesian version of the TB-Stigma Scale [38]. The Scale consisted of two forms: Form A (Patient Perspective) with 11 questions across three domains (disclosure, isolation, and guilty); and Form B (Community Perspective) with 10 questions across two domains (isolation and distancing). The tool was reliable—with a Cronbach's alpha of 0.738 for Form A (Patient Perspective) and 0.807 for Form B (Community Perspective) [38]. Aligned with the original Van Rie Stigma Score [14], from which the Indonesian score was adapted, each item had four potential answer options: strongly disagree (1), disagree (2), agree (3), and strongly agree (4). TB-Stigma scores were calculated following the original tool's guidance of the: (sum of item scores x 50) / (3 x number of item). In addition, in line with existing TB stigma measurement guidance and to enhance interpretation and potential future policy dialogue [7, 39, 40], we also presented scores by cohort quartiles: no stigma (no stigmatisation in all items), low (TB-Stigma score ≤16.67), moderate (16.68–33.33), and high TB-Stigma (>33.33). (**S1 Appendix**).

In this study, we assessed the association between TB-Stigma, depression, and QoL because these elements often intersect [20, 41, 42], therefore may be better addressed in a multi-faceted, complex, or integrated psychosocial intervention instead of a uni-faceted intervention focused on a single element.

### Patient Health Questionnaire-9 (PHQ-9)

We used the Indonesian-validated version of the Patient Health Questionnaire-9 (PHQ-9) [43], with a Cronbach's alpha of 0.837 [38], to assess the participants' symptoms of depression according to the pre-existing PHQ-9 categories: no depression (score of 0–4), mild (5–9), moderate (10–14), moderately severe (15–19), and severe (20–27) [41]. The PHQ also allows the interpretation of "major depressive disorder" (MDD) and "other depressive disorder". MDD is defined when a person indicates having more than five symptoms in PHQ9 more than half the days and *Symptom 1* (Little interest or pleasure in doing things) or *Symptom 2* (Feeling down, depressed, or hopeless) more than half the days. Other depressive disorder is defined when a person indicates having 2–4 symptoms in PHQ9 and *Symptom 1* or *Symptom 2* more than half the days. (**S2 Appendix**).

### EQ-5D-5L

QoL was measured using the Indonesian version of EQ-5D-5L [44, 45] consisting of five locally-weighted dimensions: mobility, self-care, usual activities, pain/discomfort, and anxiety/depression with five levels of severity [44]. This tool has been shown to be reliable in other studies with a Gwet's test and retest agreement coefficient of 0.85–0.99 and percentage agreement of 90–99% [44]. The assessment conformed with EuroQol guidance on QoL measurement and was locally-appropriate using the Indonesian value set of EQ-5D-5L [45].

## Psychosocial support questions

We added ten questions to measure participants' receipt of and/or perceived unmet need for psychosocial support. The study team designed the initial questions related to needs and unmet needs for psychosocial support. Then, questions were shared with an expert panel including a pulmonologist, a psychologist, community medicine specialist, national TB program staff, and a member of a TB civil society organization. The panel evaluated the questions for content validity and iteratively refined the questions [38]. Questions were divided into three recognized domains: informational, emotional, and instrumental support [46]. (**S3 Appendix**). Informational support refers to any facts, advice, or other educational materials provided to people with TB to help them solve problems. Emotional support refers to care, encouragement, and empathy to garner a sense of security amongst TB-affected people. Informational support in this study was defined as a tangible help in practical form provided through material assistance or practical program, such as counselling or meeting. This question was to assess whether and what support was currently available to them. Second, we asked whether they perceived that they needed such psychosocial support, regardless of whether they received it or not. We then assessed whether they had "unmet need" for psychosocial support, which was defined as a perceived need but lack of receipt of psychosocial support.

We included several covariates in the analysis which have previously been demonstrated to be associated with TB stigma, depression, or quality of life in other studies [7, 8], which were age, sex (male/female), participant group (A, B, C, D), area (urban/rural), job loss due to TB (yes/no), and formal education level (no education/ elementary/high school/university).

## Data collection

All psychosocial impact questions described above were combined in a single tool that could be completed with a participant in an interview of less than 15 minutes. To ensure the proper use of the tool, we recruited and trained interviewers with a background in health sciences in a one-day online training. The training included: a comprehensive explanation of the study background, rationale, methods, and participant selection; and a detailed review of the paper-based TB-Stigma tool, protocol, and Standardized Operating Procedure. Interviewers were trained to enter the data into the study's electronic database using the RedCap platform (https://redcap.fk.ui.ac.id). After data checking, cleaning, and validation, the data was analyzed using IBM SPSS Version 25.0.

## Data analysis

We applied descriptive analysis to capture TB-Stigma among people with TB and their receipt of and/or need for psychosocial support. TB-Stigma scores were summarized by means with standard deviation (SD) and range. TB-Stigma tool items from patient and community perspectives and receipt of and/or need for psychosocial support were displayed in numbers (n/N) and percentages (%). We identified clinical, health system, and socioeconomic factors associated with TB-Stigma using a general linear model (GLM) to estimate B values and their adjusted Bs (aBs). The B values were adjusted with presumed associated factors mentioned above [7, 8].

After reviewing literature, with special reference to the conceptual framework in Chen et al [19], we designed the model to look at the interaction between TB Stigma, depression, and QoL. The correlations between TB-Stigma and depression scores were evaluated by Spearman correlation tests to obtain their coefficient (R$s$). We analyzed the association between TB-Stigma scores, depression, and QoL using multinomial regression (for depression symptoms), binary logistic regression (for MDD and other depressive disorders) and GLM (for

QoL) to obtain the crude odds ratios (cORs), adjusted ORs (aORs), cBs and aBs and their 95% confidence intervals (95% CIs).

### Ethical considerations

This study received a support letter from the Indonesia's Ministry of Health, a research ethical approval from the Ethics Committee of the Faculty of Medicine, University of Indonesia (No. KET-60/UN2.F1/ETIK/PPM.00.02/2022, on January 17, 2022), and research permit from seven provincial authorities prior the implementation. All healthcare facilities involved in this study granted the permit and helped communicate with the potential participants. We provide a complete explanation to participants before they signed informed consent to participate in the study. Participants were allowed to withdraw their participation from this study at anytime without any consequences.

### Patient and public involvement

This study involved patients and public in the study design, instrument development, and data collection. The research question and the development of tools to measure TB-Stigma was informed by experts from various background, including staff from the Ministry of Health, the NTP at the province and district levels, and civil organisation, which addressed patients' experience and preferences. This study was supported by the Ministry of Health, approved by local authorities, and involved with healthcare workers in public and private healthcare facilities. The respondents of survey were people with TB who were diagnosed with TB or received TB treatment. The patient and public involvement in this study continued throughout the research process and would be followed up in TB-CAPS Study, in which we develop a peer-led, community-based psychosocial support for people with TB. The study results will be disseminated to public through public seminar and scientific workshop.

## Results

We recruited and completed interviews with 612 adults with TB and all recorded interviews were included in the analysis (**Table 1**). Most participants were male, married, lived in an urban area, and received treatment at public healthcare facilities. Of 383 participants who had income-earning jobs before TB diagnosis, 59 (15.4%) participants lost their job and 8 (2.1%) changed their job because of TB.

### TB stigma scores

This study found that 60.6% (96%CI 59.6–64.5%) experienced moderate TB-Stigma. The average TB-Stigma score in Form A (Patient Perspective) was 19.0 (SD 6.9; min-max 0–50). Most participants (60.6%) experienced moderate TB-Stigma, which was predominantly related to feelings of guilt about having TB including: being a burden on their family (45.1%), having lifestyle behaviour that participants perceived to have contributed to their TB disease such as smoking tobacco or drinking alcohol (37.4%), and worrying about having HIV/AIDS (34.0%) (**Table 2**). A high proportion of participants reported fear of disclosure of their TB disease (18.0–49.2%) and felt hurt by others' reactions to their TB disease (25.2%).

The average TB-Stigma score in Form B (Community Perspective) was 23.4 (SD 8.4, min-max 0–50). Most participants (63.9%) reported moderate perceived TB-Stigma from their community, particularly the perception of being isolated from and by people in their community (38.2–66.7%).

**Table 1. Participants' characteristics (N = 612).**

| Characteristics | | n | % |
|---|---|---|---|
| Sex | | | |
| | Male | 379 | 61.9 |
| | Female | 233 | 38.1 |
| Age, *years old*, median (IQR) | | 42 | (29–55) |
| Age, *years old* | | | |
| | 18–30 | 166 | 27.1 |
| | 31–40 | 114 | 18.6 |
| | 41–50 | 124 | 20.3 |
| | 51–60 | 108 | 17.6 |
| | >60 | 100 | 16.3 |
| Marital status | | | |
| | Not married | 140 | 22.9 |
| | Married | 431 | 70.4 |
| | Widowed | 41 | 6.7 |
| Highest educational level attained[a] | | | |
| | No schooling | 20 | 3.3 |
| | Elementary to junior high school | 187 | 30.6 |
| | Senior high school | 297 | 48.5 |
| | College/University | 108 | 17.6 |
| Having income earning job before TB diagnosis | | | |
| | Yes | 383 | 62.6 |
| | No and of working age[b] | 172 | 28.1 |
| | No and not of working age[b] | 57 | 9.3 |
| Impact of TB on employment, n = 383 | | | |
| | No impact | 272 | 71.0 |
| | Job loss or negative change because of TB[c] | 67 | 17.5 |
| | Job loss or negative change not because of TB[c] | 44 | 11.5 |
| Provinces | | | |
| | Jambi | 93 | 15.2 |
| | West Sumatera | 110 | 18.0 |
| | Jakarta | 73 | 11.9 |
| | West Kalimantan | 109 | 17.8 |
| | Bali | 36 | 5.9 |
| | South Sulawesi | 107 | 17.5 |
| | Maluku | 84 | 13.7 |
| Area | | | |
| | Urban | 414 | 67.6 |
| | Rural | 198 | 32.4 |
| People with TB receiving treatment at | | | |
| | *Group A*: Intensive phase at a public facility | 404 | 66.0 |
| | *Group B*: Private facilities | 103 | 16.8 |
| | *Group C*: LTFU to treatment[d] | 51 | 8.4 |
| | *Group D*: TB retreatment | 54 | 8.8 |

[a]Usual age of children: 6–15 years (elementary to junior high school), 15–18 years (senior high school), and >18 years (college/university)

[b]Working age in this study was defined as aged less than 60 years (pension age)

[c]Negative change refers to a decrease in income or perceived lower level of job/role

[d]LTFU to treatment: lost-to-follow-up to treatment, is those diagnosed with TB but not following up with TB treatment.

**Table 2. TB-Stigma from patient and community perspectives (N = 612).**

| Form A: Patient Perspective | n | (%) |
|---|---|---|
| *TB-Stigma Category* | | |
| No Stigma | 78 | (12.8) |
| Low | 153 | (25.0) |
| Moderate | 371 | (60.6) |
| High | 10 | (1.6) |
| *Domain*: Disclosure | | |
| P6. I am afraid to tell people outside my family that I have TB | 208 | (34.0) |
| P7. I am afraid to tell others that I have TB because others may think that I also have HIV/AIDS | 169 | (27.6) |
| P9. I choose carefully who I tell about having TB | 301 | (49.2) |
| P12. I am afraid of other people to tell my family that I have TB | 110 | (18.0) |
| *Domain*: Isolation | | |
| P1. I feel hurt by how others react to knowing that I have TB | 154 | (25.2) |
| P2. I have lost friends when I shared with them that I have TB | 75 | (12.3) |
| P3. I feel lonely | 97 | (15.8) |
| P5. I am afraid of going to TB clinics because other people may see me there | 90 | (14.7) |
| *Domain*: Guilty | | |
| P8. I feel guilty because my family has the burden of caring for me | 276 | (45.1) |
| P10. I feel guilty for getting TB because of my smoking, drinking, or other lifestyle behaviours | 229 | (37.4) |
| P11. I am worried about having HIV/AIDS | 208 | (34.0) |
| **Form B: Community Perspective** | n | (%) |
| *TB-Stigma Category* | | |
| No Stigma | 113 | (18.5) |
| Low | 59 | (9.6) |
| Moderate | 391 | (63.9) |
| High | 49 | (8.0) |
| *Domain*: Isolation | | |
| C13. Some people may not want to eat or drink with friends who have TB | 364 | (59.5) |
| C14. Some people feel uncomfortable about being near those with TB | 288 | (47.1) |
| C15. If a person has TB, some community members will behave differently towards that person for the rest of his/her life have HIV/AIDS | 234 | (38.2) |
| C16. Some people do not want those with TB playing with their children | 355 | (58.0) |
| C17. Some people keep their distance from people with TB | 408 | (66.7) |
| C22. Some people may not want to eat or drink with relatives who have TB | 333 | (54.4) |
| *Domain*: Distancing | | |
| C18. Some people think that those with TB are disgusting | 145 | (23.7) |
| C19. Some people do not want to talk to others with TB | 165 | (27.0) |
| C20. Some people are afraid of those with TB | 239 | (39.1) |
| C21. Some people try not to touch others with TB | 184 | (30.1) |

## Factors associated with TB stigma

Form A (Patient Perspective) TB-Stigma scores were also higher among males (aB 1.73; 0.59–2.87), those losing or changing job because of TB (aB 2.09; 0.31–3.88), and those living in a rural area (aB 1.41; 0.19–2.63) (**Table 3**). These Patient Perspective scores were 2.48 and 2.86 points higher among people receiving treatment at private facilities (*Group B*) and LTFU (*Group C*), respectively, than people receiving TB treatment at public facilities (*Group A*). Form A (Community Perspective) TB-Stigma scores were 1.95 and 3.28 points higher among people receiving treatment at private facilities (*Group B*) and LTFU (*Group C*), respectively,

**Table 3. Factors associated with TB-Stigma Scores from Patient (Form A) and Community (Form B) perspectives.**

| Variables | | n | Mean | (SD) | cB | 95%CI | aB | 95% CI |
|---|---|---|---|---|---|---|---|---|
| **A. Patient Perspective** | | | | | | | | |
| Group | | | | | | | | |
| | A: Treatment at public facility | 404 | 18.6 | (7.1) | | REF | | REF |
| | B: Treatment at private facilities | 103 | 20.3 | (6.0) | 1.74 | 0.24 to 3.24 | 2.48 | 0.94 to 4.03 |
| | C: LTFU to TB treatment | 51 | 21.1 | (7.9) | 2.51 | 0.49 to 4.53 | 2.86 | 0.85 to 4.87 |
| | D: Retreatment | 54 | 18.1 | (6.2) | -0.48 | -2.45 to 1.49 | -0.26 | -2.21 to 1.69 |
| Area | | | | | | | | |
| | Urban | 414 | 18.7 | (7.0) | | REF | | REF |
| | Rural | 198 | 19.8 | (6.8) | 1.10 | 0.08 to 2.28 | 1.41 | 0.19 to 2.63 |
| Sex | | | | | | | | |
| | Female | 233 | 17.9 | (6.4) | | REF | | REF |
| | Male | 379 | 19.7 | (7.2) | 1.77 | 0.64 to 2.90 | 1.73 | 0.59 to 2.87 |
| Job loss or change because of TB | | | | | | | | |
| | No | 545 | 18.7 | (6.9) | | REF | | REF |
| | Yes | 67 | 21.6 | (7.4) | 2.85 | 1.09 to 4.61 | 2.09 | 0.31 to 3.88 |
| Formal education | | | | | | | | |
| | No school | 20 | 20.8 | (8.1) | | REF | | REF |
| | Elementary | 187 | 18.8 | (7.4) | -1.94 | -5.17 to 1.28 | -2.57 | -5.75 to 0.62 |
| | High school | 297 | 19.2 | (6.6) | -1.57 | -4.74 to 1.59 | -2.50 | -5.75 to 0.76 |
| | University or college | 108 | 18.7 | (7.0) | -2.06 | -5.39 to 1.28 | -2.83 | -6.26 to 0.61 |
| Age, *years*, mean (SD) | | 612 | 19.04 | (6.9) | -0.03 | -0.06 to 0.01 | -0.05 | -0.09 to -0.01 |
| **B. Community Perspective** | | | | | | | | |
| Group | | | | | | | | |
| | A: Treatment at public facility | 404 | 22.9 | (8.0) | | REF | | REF |
| | B: Treatment at private facilities | 103 | 24.2 | (7.7) | 1.24 | -0.56 to 3.05 | 1.95 | 0.08 to 3.83 |
| | C: LTFU to TB treatment | 51 | 26.2 | (10.8) | 3.31 | 0.88 to 5.74 | 3.82 | 1.38 to 6.26 |
| | D: Retreatment | 54 | 22.3 | (9.0) | -0.65 | -3.02 to 1.72 | -0.22 | -2.59 to 2.15 |
| Area | | | | | | | | |
| | Urban | 414 | 23.3 | (8.3) | | REF | | REF |
| | Rural | 198 | 23.5 | (8.6) | 0.25 | -1.18 to 1.67 | 0.69 | -0.80 to 2.16 |
| Sex | | | | | | | | |
| | Female | 233 | 22.6 | (8.0) | | REF | | REF |
| | Male | 379 | 23.8 | (8.6) | 1.23 | -0.13 to 2.60 | 1.35 | -0.04 to 2.73 |
| Job loss or change because of TB | | | | | | | | |
| | No | 545 | 23.2 | (8.3) | | REF | | REF |
| | Yes | 67 | 24.5 | (9.0) | 1.28 | -0.85 to 3.41 | 0.62 | -1.54 to 2.79 |
| Formal education | | | | | | | | |
| | No school | | | | | REF | | REF |
| | Elementary | 187 | 22.8 | (8.6) | -1.14 | -5.00 to 2.72 | -2.28 | -6.15 to 1.59 |
| | High school | 297 | 24.1 | (8.4) | 0.18 | -3.61 to 3.97 | -1.66 | -5.61 to 2.30 |
| | University or college | 108 | 22.2 | (7.5) | -1.68 | -5.67 to 2.32 | -3.36 | -7.53 to 0.81 |
| Age, *years*, mean (SD) | | 612 | 23.4 | (8.4) | -0.05 | -0.10 to -0.01 | -0.07 | -0.12 to -0.03 |

than people receiving TB treatment at public facilities (*Group A*). People of older age had lower TB-stigma scores from both the Patient and Community perspectives (Table 3). The models were fit (R-Square = 0.066, F = 4.238, p<0.001 for patient perspective; R-square = 0.046, F = 2.887, p = 0.002 for community perspective).

**Table 4. Multivariable multinominal regression model of the association of TB-stigma and levels of depression symptoms.**

| TB-Stigma | No Depression (n = 358) | Mild-moderate (n = 220) | | | Moderately severe to severe (n = 34) | | |
|---|---|---|---|---|---|---|---|
| | mean (SD) | mean (SD) | OR; 95%CI | aOR; 95%CI[a] | mean (SD) | OR; 95%CI | aOR; 95%CI[a] |
| *Patient perspective* | | | | | | | |
| Total score | 17.50 (6.48) | 20.47 (6.58) | 1.07; 1.04–1.10 | 1.08; 1.05–1.11 | 25.94 (8.34) | 1.22; 1.15–1.30 | 1.23; 1.15–1.32 |
| Disclosure | 18.68 (9.11) | 21.59 (9.48) | 1.03; 1.02–1.05 | 1.04; 1.02–1.06 | 28.06 (11.93) | 1.11; 1.07–1.15 | 1.11; 1.06–1.15 |
| Isolation | 14.56 (7.40) | 17.67 (8.55) | 1.05; 1.03–1.08 | 1.06; 1.03–1.08 | 23.41 (9.40) | 1.14; 1.09–1.19 | 1.16; 1.10–1.23 |
| Guilty | 19.86 (9.04) | 22.70 (9.33) | 1.03; 1.02–1.05 | 1.04; 1.02–1.07 | 26.47 (10.06) | 1.08; 1.04–1.13 | 1.09; 1.05–1.14 |
| *Community perspective* | | | | | | | |
| Total score | 21.69 (7.77) | 25.33 (8.44) | 1.06; 1.04–1.08 | 1.06; 1.04–1.09 | 28.28 (9.42) | 1.10; 1.06–1.15 | 1.11; 1.06–1.17 |
| Isolation | 24.06 (8.36) | 27.89 (8.53) | 1.06; 1.03–1.08 | 1.06; 1.04–1.08 | 31.29 (9.73) | 1.11; 1.06–1.16 | 1.11; 1.06–1.17 |
| Distancing | 18.12 (9.02) | 21.48 (10.96) | 1.04; 1.02–1.05 | 1.04; 1.02–1.06 | 23.77 (10.64) | 1.06; 1.02–1.10 | 1.07; 1.03–1.11 |

[a]Adjusted for age, sex (male/female), group (intensive phase at public facilities/private facilities/LTFU to treatment/retreatment), area (urban/rural), job loss due to TB (yes/no), and formal education (no school/elementary/high school/university).

### TB stigma and depression

Depression was identified in 41.5% (95% CI 37.7–45.3%) participants. Of 612 participants, 220 (35.9%) had mild to moderate depression, and 34 (5.6%) had moderately severe to severe depression. TB-Stigma scores, for both Patient and Community perspectives, were significantly correlated with PHQ-9 score, with respective $r_s$ of 0.295 and 0.254. The average and median values of TB-Stigma scores were higher among those with depression than those without depression (**Table 4** and **S1 Fig**).

The model assessing correlation between TB Stigma Scores and depression, with adjustment for age, sex, *Group*, rural vs urban area, and job loss due to TB, were fit ($\chi^2$ = 96.926, p<0.001 for TB Stigma patient perspective; $\chi^2$ = 66.670, p<0.001 for TB Stigma community perspective). Every unit increase in TB-Stigma score was associated with a 4–8% increase in the odds of having mild to moderate depression symptoms and 6–22% increase in the odds of having moderately severe to severe depression symptoms compared to having no depression. Thirty-nine (6.4%) participants were identified as having MDD and 47 (7.7%) had other depressive disorder. Every unit increase in TB-Stigma score was associated with a 20% increase in the odds of having a MDD and a 7% increase in the odds of having another depressive disorder (**S1 Table**).

### TB stigma, depression, and QoL

The average QoL among adults with TB without depression was 0.9 (SD 0.13, min-max 0.28–1.00). The model assessing correlation between TB Stigma, depression, and QoL with adjustment for age, sex, *Group*, rural vs urban area, and job loss due to TB, were fit (R-square = 0.324, F = 23.878, p<0.001 for TB Stigma patient perspective; R-square = 0.104, F = 6.350, p<0.001 for TB Stigma community perspective). Stigma and depression were significantly associated with lower QoL (**Table 5**). QoL was lower among those with moderately severe to severe depression (aB -0.383; 95%CI -0.445 to -0.322) and mild to moderate depression (aB -0.155; 95%CI -0.184 to -0.126) than those without depression. Higher TB-Stigma scores correlated negatively with QoL, and this association was stronger in those with either mild-moderate or moderately severe to severe depression. (**Table 5**). While TB Stigma scores among those LTFU and those receiving treatment at private facility were higher lower than those treated at public facility, their QoL was significantly lower. (**S2 Table** and **S2 Fig**).

**Table 5. Association of TB-stigma and levels of depression symptoms with quality of life (N = 612).**

| Variables | | n | Quality of Life | | | | | |
|---|---|---|---|---|---|---|---|---|
| | | | Mean (SD) | | cB | 95%CI | aB | 95% CI |
| *Depression* | | | | | | | | |
| | No depression | 358 | 0.91 | (0.13) | | REF | | REF |
| | Mild to moderate | 220 | 0.76 | (0.21) | -0.155 | -0.185 to -0.125 | -0.155 | -0.184 to -0.126 |
| | Moderately severe to severe | 34 | 0.53 | (0.33) | -0.376 | -0.439 to -0.314 | -0.383 | -0.445 to -0.322 |
| *A. Patient Perspective* | | | | | | | | |
| Stigma | | | | | -0.007 | -0.010 to -0.005 | -0.008 | -0.01 to -0.005 |
| Stigma*depression | | | | | | | | |
| | Stigma*Mild to moderate | | | | -0.007 | -0.009 to -0.005 | -0.007 | -0.010 to -0.005 |
| | Stigma*Moderately severe to severe | | | | -0.013 | -0.015 to -0.010 | -0.013 | -0.016 to -0.010 |
| *B. Community Perspective* | | | | | | | | |
| Stigma | | | | | -0.004 | -0.006 to -0.002 | -0.004 | -0.006 to -0.002 |
| Stigma*depression | | | | | | | | |
| | Stigma*Mild to moderate | | | | -0.004 | -0.006 to -0.002 | -0.004 | -0.006 to -0.003 |
| | Stigma*Moderately severe to severe | | | | -0.010 | -0.012 to -0.007 | -0.011 | -0.013 to -0.008 |

## Needs and unmet needs of psychosocial support

Informational (20.8%), emotional (25.9%) or instrumental (10.6%) support from peer groups was limited (**Table 6**). Although the need for informational, emotional, and instrumental support from peers was reported to be lower than the need for support from other groups (family and healthcare workers), the unmet needs for such peer support was high (52.0% for emotional support, 63.0% for informational support, 76.4% for group meeting, and 77.5% for group counselling from peers). The unmet need for such supports were also consistently higher among those with higher TB-Stigma scores (**S3 Table**).

## Discussion

Our novel findings from a nationally-representative cross-sectional survey show that there is a sizeable and intersecting burden of TB-Stigma and depression experienced by adult people

**Table 6. Needs and unmet needs of psychosocial support among people with TB.**

| Psychosocial support | Received (N = 612) | | Need (N = 612) | | Unmet needs* | |
|---|---|---|---|---|---|---|
| | n | % | n | % | n/N | % |
| *Information support* | | | | | | |
| TB information for myself by HCW | 523 | 85.5 | 571 | 93.3 | 56/571 | 9.8 |
| TB information for my family members by HCW | 463 | 75.7 | 545 | 89.1 | 91/545 | 16.7 |
| TB information in a peer group meeting | 127 | 20.8 | 324 | 52.9 | 204/324 | 63.0 |
| *Emotional support* | | | | | | |
| Emotional support from HCW | 535 | 87.4 | 565 | 92.3 | 38/565 | 6.7 |
| Emotional support from family | 565 | 92.3 | 577 | 94.3 | 18/577 | 3.1 |
| Emotional support from peer | 159 | 25.9 | 325 | 53.1 | 172/325 | 52.9 |
| *Instrumental support* | | | | | | |
| Home visit by HCW | 314 | 51.3 | 419 | 68.5 | 135/419 | 32.2 |
| Peer group meeting | 65 | 10.6 | 237 | 38.7 | 181/237 | 76.4 |
| Individual counselling | 93 | 15.2 | 262 | 42.8 | 172/262 | 65.6 |
| Group counselling | 58 | 9.5 | 231 | 37.7 | 179/231 | 77.5 |

*HCW,* healthcare workers *Unmet needs are the proportion of those who need the support but did not receive it.

with TB in Indonesia, which negatively impacts on QoL. TB-Stigma scores were higher among males, younger people, people receiving TB treatment at private healthcare facilities, and those who lost or changed job due to TB. There were also high levels of unmet need for psychosocial support, especially related to peer-group activities and interventions.

The high levels of TB-Stigma experienced by people with TB in Indonesia related particularly to a reluctance to disclose to others about their disease and feeling guilty about having TB. Although the TB-Stigma scores in this study were lower than those found amongst people with TB in Thailand [14] and Cambodia [47], they were higher than those found amongst people with TB in Vietnam [48] and another Indonesian report that did not use a locally-validated stigma measurement tool [49]. Our findings suggests that many people with TB would choose carefully whom they inform about their disease and are afraid of telling people outside of their family. This result conforms with the participants' anticipation of behaviours from their community including keeping their distance from, and isolating, people with TB. Such TB-Stigma perceptions may stem from fear of being "blamed as the source of disease" in their neighbourhood [25], perceived as being disgusting or dirty [47, 50], losing social status [21], receiving verbal abuse or being the subject of gossip, [50] or associated decreased employment opportunities and job loss [12, 47], much of which stems from misunderstanding and/or erroneous health beliefs concerning TB [6, 7, 51].

TB-Stigma was higher among those receiving TB treatment at private facilities or those LTFU. Rather than being determinants of TB-stigma, these are likely to be consequences of TB-stigma experienced by these specific groups. People who have concerns about TB-stigma with relation to the visibility of attending the most commonly used public care facilities in their community are more likely to use private facilities initially or switch to private facilities during the course of their healthcare seeking journey [52, 53]. It is critical to explore the perceptions of people with TB symptoms about their preferences for optimal treatment facilities and services to reduce TB-stigma. Importantly, there has been evidence to suggest that private care providers have been reluctant to offer TB services due to TB-Stigma, including against healthcare workers who care for people with TB, and service fragmentation [54]. Given the high preference for seeking care at private facilities in Indonesia, particularly in a rural areas [30, 54], there is a pressing urgency to improve the quality of TB care delivered by private providers to increase treatment uptake and completion [55]. It is also vital to improve health- and TB-related education in both the private and public sector because improving people with TB's knowledge about the disease and the importance of TB treatment has been shown to reduce LTFU [6, 7].

The economic consequences of TB may play a significant role in TB-Stigma. This study found that TB-Stigma scores were higher among males, younger people of the most economically productive age, and those experiencing job loss or decreased economic productivity because of TB. Most primary income earners within households in LMICs, including Indonesia, are males. Related research from LMICs has shown that males are concerned by TB and related TB-stigma because they may collectively corrode their primary income earner status and hence their value and status more broadly within the household and community [47, 52, 56–58]. Improving knowledge about TB at community level, reducing misperceptions about TB infectiousness and transmission within workplaces, and strengthening social protection and employment legislation could all contribute to preventing unnecessary job loss or unnecessarily prolonged time unable to work [5]. Moreover, gender and gendered norms, including perceptions of masculinity, should be considered in the design and delivery of psychosocial and economic interventions for people, and especially men, with TB [59].

This study underlines the intersectionality of TB-stigma, depression and QoL. In addition to biological factors inducing depression among people with TB [60], TB-Stigma is among the

psychosocial factors contributing to depression, which is the most common comorbid mental health disorder in people with TB [42, 61]. Depression and TB-Stigma can occur from symptom onset, including relating to weight loss and changes in appearance and body habitus, and lead to delay in care seeking, diagnostic, and treatment initiation [61]. As suggested by our findings, TB-Stigma and depression experienced after TB treatment initiation may be associated with difficulties in adhering to TB treatment, LTFU, and impaired quality of life [20, 41, 62]. Hence, syndemic TB, TB-Stigma, and depression have the potential to act synergistically and thereby impact negatively upon QoL and levels of disability during and beyond TB illness.

Our findings support the use of routine screening for both TB-Stigma and depression as part of a holistic approach to addressing the needs of people with TB. To do this, linkage of TB care with mental health services, which has only been implemented in a handful of countries to date [63], could be critical. In LMICs, where specialist mental health services are often lacking, an integrative approach should facilitate the provision of mental health counselling in primary care. Training non-specialist healthcare workers at the primary care level will be essential to provide low cost and potentially cost-effective interventions to deliver mental health screening, triage, management, and prompt referral [64, 65]. Psychological counselling could act as a gateway to onward referral to other psychosocial intervention activities such as peer-led mutual support groups or "TB Clubs" [66].

Peer group support is an essential psychosocial intervention. Despite the increasing awareness of the importance of providing peer support to people with TB, to date this has been piecemeal, difficult to replicate, and had neither rigorous process nor effectiveness evaluation. Indeed, peer support for people with TB has been less frequently provided than for people living with HIV/AIDS, in which such groups are well established [42, 63]. We found that the reported need for peer group support was lower than support through other mechanisms such as home visits and individual counselling. This may relate to the TB-stigma score findings specifically relating to fear of disclosure and a lack of trust in others, especially outside of the household, some of which may be due to feelings of isolation and marginalization from the surrounding community or, if being retreated, previous experiences during TB illness and care-seeking. This supposition appears to be supported by the finding of high unmet needs for peer group meetings and group counselling, which was highest amongst those experiencing higher levels of TB-Stigma. One study has identified cultural and religious belief approaches that act as a health communication channel to reduce TB-Stigma, which may be affected by peer solidarity. Developing a more robust conceptualization, design, and delivery of peer group support (including both meetings and mutual-support counselling sessions) with a standardized package, effective training, and structured evaluation of feasibility, acceptability, effectiveness, and cost-effectiveness will be critical to consider whether scale-up and integration into routine TB services is locally appropriate.

This study has several limitations. First, Indonesia has diverse geography and cultures, which may affect the perceptions of TB-Stigma, depression, and the need for social support. We recruited 612 participants across seven purposively selected study provinces of Indonesia because they represented areas with diverse TB burdens, geographical topography, and socio-cultural norms. Nevertheless, our results may not be generalisable to Indonesia at national level or the SEARO region more broadly. Second, we only captured TB-Stigma, depression, and the need for social support among people with drug-susceptible TB and ruled out those with DR-TB, who have been found in other settings to be at a higher risk of TB-Stigma and depression. Third, using PHQ-9, we measured only depression, which may have underestimated the prevalence of other manifestations of mental illness, including anxiety, and their association with TB-Stigma. Fourth, the grouping of TB-Stigma Scale (no stigmatisation, low, moderate, and high) using quartile cut-off was not previously validated. The grouping would

help practical policy interpretation, but any future studies need to validate the cut-off for better interpretation.

In Indonesia, people with TB have high rates of intersecting TB-Stigma and depression. Being male, of younger working age, receiving TB treatment at private healthcare facilities, and losing work due to TB were the factors most strongly associated with higher TB-stigma levels. There was a large unmet need for social support through peer-led mutual support groups. These findings emphasise the importance of using locally-validated tools to measure the psychosocial impact of TB. Evidence-based, standardized but adaptable training and intervention packages, including peer-led mutual support groups, should be developed to defray the psychosocial impact of TB by reducing TB-Stigma and depression, and improving QoL among people with TB.

## Supporting information

**S1 Checklist. STROBE checklist.**
(DOC)

**S1 Appendix. Culturally adapted and validated TB-Stigma scale.**
(DOCX)

**S2 Appendix. Patient Health Questionnaire-9 (PHQ-9).**
(DOCX)

**S3 Appendix. Instrument to measure social support received and needed by people with TB.**
(DOCX)

**S1 Fig. Correlation between TB-Stigma and PHQ scores, and TB-Stigma score between depression groups.** White horizontal lines in the boxplot are the median values.
(DOCX)

**S2 Fig. TB-Stigma, depression, and quality of life between groups.**
(DOCX)

**S1 Table. TB-Stigma and depression symptoms.**
(DOCX)

**S2 Table. TB-Stigma, depression, and quality of life between groups.**
(DOCX)

**S3 Table. Unmet need of psychosocial support and TB-Stigma.**
(DOCX)

## Acknowledgments

We acknowledge the supports from all enumerators (Annisa Melianriza, Rima Moehira [West Sumatera]; Fauzan Imari, Aprilya Elchamonika, Rania Nabila Balkis [Jambi]; Salsabila Auni Putri, Rossa Maulida Falatehan [Jakarta]; Prihan Fakhri [West Kalimantan]; IDGA Narendra Suputra, Ni Kadek Putri Ayu Aprilia Swandewi, Ni Wayan Hilda Yani, I Made Tejamurti Anggara, Ida Ayu Made Gia Cahyani [Bali]; Mardiana, Jumriana, Husnul Khotimah, Dian Nur Alisah [South Sulawesi]; and Saada Lestaluhu, Mahvut T, Sakina A. Tehuayo, Sitti Johri Nasela, Samsia Rumuar, Rafela Suarlembit, Rahma Abdurahman Suatkab [Maluku]), province-, district-, and Puskesmas-level TB officers in seven provinces.

## Author Contributions

**Conceptualization:** Ahmad Fuady, Muchtaruddin Mansyur, Tom Wingfield.

**Data curation:** Ahmad Fuady, Bustanul Arifin, Ferdiana Yunita, Saidah Rauf, Agus Fitriangga, Agus Sugiharto, Finny Fitry Yani, Helmi Suryani Nasution, I. Wayan Gede Artawan Eka Putra.

**Formal analysis:** Ahmad Fuady, Ferdiana Yunita.

**Funding acquisition:** Ahmad Fuady, Agus Sugiharto, Tom Wingfield.

**Investigation:** Bustanul Arifin, Ferdiana Yunita, Saidah Rauf, Agus Fitriangga, Finny Fitry Yani, Helmi Suryani Nasution, I. Wayan Gede Artawan Eka Putra.

**Methodology:** Ahmad Fuady, Tom Wingfield.

**Project administration:** Ferdiana Yunita, Saidah Rauf, Agus Fitriangga, Agus Sugiharto.

**Resources:** Ahmad Fuady, Finny Fitry Yani, Helmi Suryani Nasution, I. Wayan Gede Artawan Eka Putra.

**Software:** Ahmad Fuady.

**Supervision:** Ahmad Fuady, Muchtaruddin Mansyur, Tom Wingfield.

**Validation:** Ahmad Fuady, Muchtaruddin Mansyur, Tom Wingfield.

**Visualization:** Ahmad Fuady.

**Writing – original draft:** Ahmad Fuady.

**Writing – review & editing:** Ahmad Fuady, Bustanul Arifin, Ferdiana Yunita, Saidah Rauf, Agus Fitriangga, Agus Sugiharto, Finny Fitry Yani, Helmi Suryani Nasution, I. Wayan Gede Artawan Eka Putra, Muchtaruddin Mansyur, Tom Wingfield.

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
