## [Decision Letter · Decision Letter 0]

3 Jul 2023

PGPH-D-23-00810

Stigma, depression, quality of life, and the need for psychosocial support among people with tuberculosis in Indonesia: a multi-site cross-sectional study

Dear Dr. Wingfield,

Thank you for submitting your manuscript to PLOS Global Public Health. After careful consideration, we feel that it has merit but does not fully meet PLOS Global Public Health’s publication criteria as it currently stands. Therefore, we invite you to submit a revised version of the manuscript that addresses the points raised during the review process.

We look forward to receiving your revised manuscript.

Kind regards,

Anil Gumber, Ph.D.

Academic Editor

Journal Requirements:

1. Please include a complete copy of PLOS’ questionnaire on inclusivity in global research in your revised manuscript. Our policy for research in this area aims to improve transparency in the reporting of research performed outside of researchers’ own country or community. The policy applies to researchers who have travelled to a different country to conduct research, research with Indigenous populations or their lands, and research on cultural artefacts. The questionnaire can also be requested at the journal’s discretion for any other submissions, even if these conditions are not met.  Please find more information on the policy and a link to download a blank copy of the questionnaire here: https://journals.plos.org/globalpublichealth/s/best-practices-in-research-reporting. Please upload a completed version of your questionnaire as Supporting Information when you resubmit your manuscript.”

2. Our staff editors have determined that your manuscript is likely within the scope of our Global Mental Health: challenges, opportunities, and the future of the field. This editorial initiative is headed by a team of Guest Editors for PLOS GPH: Rochelle Burgess (University College of London) and Dixon Chibanda (University of Zimbabwe and London School of Tropical Medicine and Hygiene). The Collection invites researchers to submit original research which engages with, or disrupts, the urgent needs across the global mental health landscape. We especially encourage submissions of studies that critically interrogate the status quo of the field and that involve inter-/trans-disciplinary approaches and those which share perspectives from underrepresented global regions and communities.

 Additional information can be found on our announcement page: https://collections.plos.org/call-for-papers/global-mental-health-opportunities-challenges/ 

If you would like your manuscript to be considered for this collection, please let us know in your cover letter and we will ensure that your paper is treated as if you were responding to this call.  Please note that being considered for the Collection does not require additional peer review beyond the journal’s standard process and will not delay the publication of your manuscript if it is accepted by PLOS GPH. If you would prefer to remove your manuscript from collection consideration, please specify this in the cover letter.

3. Please provide separate figure files in .tif or .eps format.

Additional Editor Comments (if provided):

Some additional Analysis is required by Four Groups of Patients. Most tables including depression outcomes need to be tabulated by Four Groups. I am surprised the data on EQ5D is collected but analysis is not done for each of five dimensions across four groups. Also I am unable to see analysis based on QALYs score.

Reviewers' comments:

Reviewer's Responses to Questions

**Comments to the Author**

1. Does this manuscript meet PLOS Global Public Health’s publication criteria? Is the manuscript technically sound, and do the data support the conclusions? The manuscript must describe methodologically and ethically rigorous research with conclusions that are appropriately drawn based on the data presented.

Reviewer #1: Partly

Reviewer #2: Yes

2. Has the statistical analysis been performed appropriately and rigorously?

Reviewer #1: No

Reviewer #2: Yes

3. Have the authors made all data underlying the findings in their manuscript fully available (please refer to the Data Availability Statement at the start of the manuscript PDF file)?

Reviewer #1: Yes

Reviewer #2: Yes

4. Is the manuscript presented in an intelligible fashion and written in standard English?

Reviewer #1: Yes

Reviewer #2: Yes

5. Review Comments to the Author

Reviewer #1: In the introduction, make sure that the objective and research question is clearly presented.

Author should use sub-headings for the tools that were used to measure stigma, depression, QoL, etc.

Report the reliability and internal consistencies of the tools in this study.

Outcome and primary exposure covariates needed to be clearly mentioned in the method section separately.

Author should use the STOBE reporting guideline for cross sectional study.

Require a section on how the questionnaire was created and on which page what kind of information was collected. How the questionnaire was validated and piloted.

“Ten questions were added to measure participants’ receipt of and/or perceived unmet need for psychosocial support.” Was it a valid tool like PHQ-9? If not, how was this tool validated? Internal consistencies require reporting.

In the statistical analysis, the raised research question in the introduction should be answered.

Was the assumption of the models met? Authors should report all the assumptions of all models that they used and how the model was built, such as variable selection, check of correction, interaction/effect modification, and confounders.

The author can not include the variables in the model without forming research questions that they are looking to answer, and variable selection should be explained with the check of correction, interaction/effect modification, and confounders.

All models should be reported their fitness.

Results should be presented in sub-headings that will portray the research questions that they are going to answer.

Reviewer #2: This is an important work on stigma, depression and quality of life of people with drug sensitive tuberculosis in Indonesia. The research is well designed with the appropriate methodology to respond to the research questions. Overall, the sidings are important to design appropriate psychosocial support for the patients.

The overall structure of the manuscript is very good with clear flow of each part to tell the story. I have the following comments:

1. In the methods part the stigma was classified as no stigma, low, moderate and high based on the quartile of the cohort. I believe this misleading, without the presences of a validated cutoff point, to divided stigma in such a way is not helpful. Even, a single experience in one of the question included in the scale might have profound impact on the patients outcome. I would suggest the authors treat stigma as a continues variable and present the mean and standard deviation across the manuscript. Accordingly the statement '60.6% experienced moderate TB-Stigma' in the abstract should be revised, please included the mean +/- SD and the range.

2. In the abstract 'depression was identified in 41.5% participants', you provided the point estimates. It would be good to included the 95%CI.

2. How Patient Health Questionnaire-9 (PHQ-9), results are interpreted is not included in the main manuscript, say what is mild or moderately severe depression means. The authors should also provide references to the sources of such classifications.

3.When presenting the mean for the different tool for the first time, it is important to indicate the range (minimum and maximum scores). This will help to put the results into context.

4.

6. PLOS authors have the option to publish the peer review history of their article (what does this mean?). If published, this will include your full peer review and any attached files.

**Do you want your identity to be public for this peer review?** For information about this choice, including consent withdrawal, please see our Privacy Policy.

Reviewer #1: No

Reviewer #2: No

<quillbot-extension-portal></quillbot-extension-portal>

---

## [Editor Report · Decision Letter 1]

28 Nov 2023

Stigma, depression, quality of life, and the need for psychosocial support among people with tuberculosis in Indonesia: a multi-site cross-sectional study

PGPH-D-23-00810R1

Dear Dr Wingfield,

We are pleased to inform you that your manuscript 'Stigma, depression, quality of life, and the need for psychosocial support among people with tuberculosis in Indonesia: a multi-site cross-sectional study' has been provisionally accepted for publication in PLOS Global Public Health.

Best regards,

Anil Gumber, Ph.D.

Academic Editor